# The Effects of the COVID-19 Pandemic on the Parenting of Infants: A Couples Study

**DOI:** 10.3390/ijerph192416883

**Published:** 2022-12-15

**Authors:** Paulina Anikiej-Wiczenbach, Maria Kaźmierczak

**Affiliations:** Institute of Psychology, University of Gdansk, 80-309 Gdansk, Poland

**Keywords:** parenting, parental responsiveness, COVID-19, perceiving a child’s cry

## Abstract

(1) Background: The COVID-19 pandemic has led to many negative changes in everyday functioning. This study aimed to establish how it impacts parental responsiveness towards their children; (2) Methods: 132 couples (N = 264) who were parents of young children (from 3 to 24 months; M = 12.61; SD = 6.71) participated in this study. The Parental Responsiveness Scale was used to measure parental responsiveness toward their own child and the Polish adaptation of the My Emotions Scale was used to measure emotional reactions to the child’s cry. We collected data about perceived stress, fear of being affected by COVID-19, and emotional overload caused by the pandemic. An analysis using actor–partner interdependence models was carried out; (3) Results: there were actor effects for both parental responsiveness and reactions to the child’s cry (for all measured aspects (frustration, amusement, anxiety, empathy, sympathy)). For women, parental responsiveness was a negative partner effect of stress, and for men, there was a positive effect of fear of being infected, emotional overload, and stress; (4) Conclusions: these results show how important it is to take care of families and investigate the effects of the pandemic on their functioning.

## 1. Introduction

The COVID-19 pandemic began in December 2019 and rapidly spread throughout the world. An increasing number of affected people and deaths has caused the deterioration of everyone’s well-being and mental health. This has affected not only people with pre-existing mental illnesses but also healthy individuals [1].

In Poland, the implementation of strategies for reducing social contact occurred from March 2020. First, it started with a complete lockdown. Then, in June, it changed to some limitations in access to public services, such as restaurants, places of entertainment (cinemas, theatres), and in the organization of public events. Everyday living changed [2]. These public limitations and precautions limited the spread of the virus, but at the same time, brought negative consequences (e.g., job losses, deterioration of well-being) and many new stressors. Among them were anxiety, social isolation, uncertainty, intensification of chronic and mental diseases, frustration, fear of the virus, panic behavior, and other consequences of being in a highly stressful situation [1,3,4]. Moreover, it has had an impact on economics and social life (e.g., changes in work roles, everyday routines, and the functioning of social-support institutions) [5]. This also influenced the functioning of the families [6].

The COVID-19 pandemic was potentially a very challenging time for parents, who may have experienced parenting stress associated with responsibility for their children’s lives [7]. Families all over the world were faced with new stressors that threatened their safety, health, and economic well-being. Indeed, a higher level of perceived parental stress during the global pandemic has been already reported [8,9]. Parents experienced a higher level of stress even if they were not exposed to the virus but though having to deal with difficulties, such as the overload of negative emotions, fear for their own lives and their partners’ lives, and those of their children [10].

Regarding individual differences in coping strategies, people could present different levels of anxiety or other psychosocial outcomes [11]. Furthermore, people can experience greater negative consequences depending on their social situation (e.g., being a primary caregiver). One of the factors that help people cope in such times may be the support provided by a romantic partner [7].

The pandemic forced parents to cope with negative outcomes such as emotional, cognitive, and even physical fatigue [12], that may have impacted their relations with their infants and children [13,14]. A relationship between stress and anxiety related to the COVID-19 pandemic and negative maternal and neonatal outcomes has been demonstrated [7]. According to one follow-up study, changes in everyday functioning had an impact on the development of anxiety and depressive symptoms in mothers of young children [15]. Fulfilling children’s needs may have become more difficult and may also have caused frustration for parents who tried to share childcare with their partner while working remotely. Furthermore, it has been suggested [16] that the pandemic may have caused parents to experience lower self-efficacy.

The pandemic has caused limited access to institutions focused on family and child needs. Increasing parental frustration and emotional overload are risk factors of child maltreatment [17], but they have been less frequently reported because of limited access to institutional support (e.g., schools, kindergarten), where these signs of maltreatment could be seen [18]. Indeed, stress may impact one’s relationship with one’s own child [19] and may even lead to harsh parenting [20]. Additionally, parents must deal with limited access to childcare centers, which help parents meet the many developmental and social needs of their children [21]; thus, some parents also had to assume the role of educator. This paper focuses on the parents of young children and their strategies for coping with stress. In the face of the pandemic, they also had to cope with adjusting to a new role [19] alongside an intensified feeling of responsibility for the health and development of their helpless child. For the child’s well-being and adequate development, they had to remain responsive to the child’s cues and respond to them sensitively and adequately [22,23]. This response should be prompt and provided with empathy and tenderness These sensitive reactions depend on situational context [24], and thus reactions to a child’s crying could be affected by the difficult situation of the global pandemic, as experiencing a high level of stress reduces one’s ability to perceive a child’s signals [7].

It should be emphasized that children communicate with parents using a wide range of different verbal and nonverbal cues, such as gaze, smile, and vocalizations. However, one of the strongest and most discomfort-inducing stimuli informing the parent about the need to is the child’s cry [25]. Reactions to the cries of one’s own child vary depending on situational context and can lead to emotional overload [26]. Parental responsiveness and parental child-oriented reactions to cries are highly connected with empathy in terms of empathic concern and perspective-taking [27]. However, a child’s cry can provoke different reactions in parents. Some focus on the child—sympathy, empathy; others focus on the parent—frustration, anxiety, and even amusement [28].

Previous studies have shown that mothers appear to be more responsive than fathers [27]. However, in the time of the pandemic, the situation may have changed because of the possibility of emotional overload to which women are more likely to be predisposed. One of the reasons could be that mothers are the ones who stay at home with the child more often. However, the relationship between parents has been shown to be crucial for the providing of the adequate emotional responses to a child. There are associations between engagement with a child and observed responsive caregiving for both fathers and mothers. The strengths of agreement and associations between parents in terms of provided stimulation were greater among couples who had higher quality coparenting relationships [29]. Thus studies of parental responsiveness and factors during the pandemic should be performed on both parents. It is crucial to study the mutual interactions between partners.

The aim of the presented research was to explore the consequences of the COVID-19 pandemic on the functioning of parents in terms of their responsiveness to their own child’s cues. We analyzed whether perceived stress, emotional overload, and fear of being affected by COVID-19 were connected with parental responsiveness, constituted by sensitivity in perceiving the child’s cues, parental willingness to react emotionally, physical availability (e.g., to comfort the baby), promptness in responding to the child’s needs, and the adequacy of parental reactions and emotions connected with child’s cry [27].

## 2. Materials and Methods

The study was conducted during the COVID-19 pandemic (from November 2020 to March 2021) in Poland. Parents were contacted through a database from previous studies (https://osf.io/xebhg/, accessed on 14 April 2021) and also via advertisement in social media. Parents who were willing to participate were contacted via e-mail. In the e-mail, the purpose of the study as well as basic information about the duration and inclusion criteria of the study (being a parent of a child aged from 3 months to 2 years) were described. The participants filled in online version of the questionnaires because of the pandemic and the high risk and discomfort of filling out paper versions in person. All research tools were in the Polish language. The completion of the set of questionnaires took about 30 min. Consent for the study was obtained at the beginning of the study (required to mark consent on the online questionnaire). Parents were asked to refer to their youngest child while responding to questions.

### 2.1. Study Group

The study group consisted of 132 couples (N = 264) parents of young children (from 3 to 24, M = 12.61; SD = 6.71). The gender distribution of the children was almost equal: 68 couples had a male child. The mothers’ ages ranged from 18 to 43 (M = 29.76; SD = 5.25) and the fathers’ ages ranged from 20 to 46 (M = 31.92; SD = 5.21). In the sample, 71.2% parents were married and the duration of their marriage ranged from 1 to 25 years (M = 4.81; SD = 3.97); 10 couples reported some problems with getting pregnant. More than half of the couples (n = 74) attended childbirth classes. The majority (n = 84) had one child, and others had 2 or 3 children. The majority of participants had higher education (n_female_ = 96; n_male_ = 74) and the others had secondary education (n_female_ = 27; n_male_ = 43), elementary education (n_female_ = 7; n_male_ = 13), or occupational education (n_female_ = 2; n_male_ = 2).

### 2.2. Research Tools

Parental responsiveness was measured by the Parental Responsiveness Scale [27]. It consists of 13 statements that pertain to parental sensitive responsiveness towards young children (up to 2 years). Parents respond on a 7-point scale, where 1 means “I totally disagree” and 7 “I definitely agree”. It is a unidimensional scale; Cronbach’s α reliability in this study was 0.83. The higher the score on this scale, the higher the declared parental responsiveness.

Emotional reactions to the child’s crying were measured with the My Emotions Scale—emotional reactions to the child’s crying [25] in the Polish adaptation [28]. Parents were asked to assess on a 5-point response scale the frequency of particular emotional reactions to their child’s cry. The instrument contains 5 subscales (each composed of 4 items) that create two main dimensions: parent-oriented reactions to the child’s cry: (1) amusement (emotions opposite to those of the child, whose emotions are ignored), (2) anxiety (helplessness, worrying about one’s own image and effectiveness as a parent), (3) frustration (irritation and blaming the child); and child-oriented reactions to the child’s cry: (1) sympathy (compassion, worrying about the child, sadness) and (2) empathy (warm emotions, willingness to protect and care). Since we used the same scale in both reported studies, we report McDonald’s ω and Cronbach’s α reliability coefficients for all the participants. All subscales had satisfactory reliability, and McDonald’s ω were: parent-oriented, 0.88, and its components were 0.90 for component 1, 0.69 for component 2, and 0.79 for component 3; child-oriented reactions, 0.96, and its components were 0.80 for component 1, 0.85 for component 2 [30].

Data about feelings regarding the COVID-19 situation were gathered through three questions with a 1 to 10 response scale. The questions were about: (1) perceived stress; (2) fear of being infected; (3) emotional overload provoked by the situation of the COVID-19 pandemic.

An Actor–Partner Interdependence Model (APIM) is a statistical method that tests interdependence in a relationship. The two major effects within APIMs are actor effects and partner effects. The actor effect is intrapersonal and is the effect of an individual’s independent variable on their own dependent variable. The partner effect is interpersonal and is the effect of a partner’s independent variable on an individual’s dependent variable. Six different actor–partner interdependence models [31] were examined, with stress, fear of being infected, and emotional overload as the independent variables in each model and the dependent variables being amusement, parent-oriented anxiety, parent-oriented frustration, child-oriented sympathy, child-oriented empathy, and parental responsiveness (see Figure 1). The APIMs were tested using the Lavaan 0.6–7 package [32] for R 4.0.3 [33] with the Robust Maximum Likelihood (MLR) estimator. All tests were two-tailed, and the significance level was set to α = 0.05. SPSS 27.0 (IBM, Armonk, NY, USA) was used to calculate the means, standard deviations, and Pearson correlation coefficients.

Additionally, two mediation analyses were performed, separately for women and for men, in which stress was the independent variable, parent-oriented frustration was the mediator, and parental responsiveness was the dependent variable. The mediation analyses were conducted using the PROCESS macro [34].

## 3. Results

The results showed that, in women, emotional overload was correlated with anxiety appearing during the pandemic. Perceived stress during the time of pandemic was correlated with frustration with the child’s cry. In men, the fear of being infected of COVID-19 virus was correlated with child-oriented sympathy during the cry. Perceived stress provoked by the pandemic was correlated with child-oriented empathy during crying. These results and the mean scores, standard deviations, and correlation coefficients are presented in Table 1.

The results of the APIM analyses are presented in Table 2.

For women’s amusement, there were no effects. For men’s amusement in response to crying, only women’s fear of being infected had a positive effect, which was on the verge of statistical significance.

Women’s and men’s emotional overload provoked by the pandemic had a positive effect on women’s parent-oriented anxiety during the child’s crying. For men’s parent-oriented anxiety, there were no effects.

Men’s stress during the pandemic and women’s fear of being infected by the virus had a positive effect on women’s parent-oriented frustration during crying. There was also a positive effect of women’s emotional overload, which weas on the verge of statistical significance. For men’s parent-oriented frustration, there was a positive actor and partner effect of emotional overload, and also a negative actor effect of fear of being infected was on the verge of statistical significance.

For women’s child-oriented sympathy, there were no effects. Men’s stress during the pandemic had a negative effect on men’s child-oriented sympathy during child crying. Furthermore, for men’s sympathy in response crying, there was a positive actor and partner effect of fear of being infected.

The negative effect of men’s stress during the pandemic on the women’s child-oriented empathy during crying was on the verge of significance. For men’s child-oriented empathy, there was a positive effect of men’s stress and fear of being infected, and a negative effect of women’s stress that was on the verge of statistical significance.

Men’s stress provoked by the COVID-19 pandemic had negative effect on women’s responsiveness. Moreover, there was a positive partner effect of emotional overload for women’s that was on the verge of statistical significance.. For men’s parental responsiveness, there was a positive effect of their own fear of being infected and of their own emotional overload. Furthermore, for fathers’ responsiveness, there was a positive effect of their own stress as well as a positive effect of women’s stress; these were on the verge of statistical significance.

Although stress and parental responsiveness were not correlated in either women or men, it has been argued that it is justified to perform mediation analysis even if path c is not significant [35]. For women, there was a significant indirect effect of stress caused by the pandemic on parental responsiveness through parent-oriented frustration during crying (based on confidence intervals), while no such effect was found for men. The results of the mediation analyses are presented in Figure 2 and Figure 3 and Table 3.

## 4. Discussion

The results show that parents’ experience of the effects of the pandemic is linked to how they look after their children. Stress caused by the pandemic situation was positively related to two different ways of reacting their child’s crying. This study showed that parents’ frustration and anxiety when they hear their baby cry increases as the perceived stress increases. Studies [36] have reported that the COVID-19 pandemic is a traumatic stress situation, and our results indicate that there are many different emotions that parents have to deal with during the pandemic. The presented study focused on changes in parental reactions towards infants caused by the pandemic.

Reporting of cases of maltreatment has decreased during the pandemic due to, for example, children not being in school; however, it is likely that maltreatment has increased. This may be associated with difficulties in coping with emotions and increased frustration [20]. However, there is evidence that stress is a risk factor for harsh parenting across the globe (due to, e.g., insufficient support in relationships, work factors, etc. [37]).

However, parental responsiveness was influenced by the pandemic situation. Parents with higher levels of fear of being affected or parents who were more emotionally overloaded by the pandemic situation and were also more responsive towards their children. This brings to mind the observation that parental responsiveness is a type of behavior strongly embedded in situational context. The threatening situation of the pandemic might lead to subjectively perceived higher parental involvement and attentiveness in responding to the child’s signals and reacting to them more promptly and adequately [27]. Moreover, our previous studies showed that parental responsiveness is linked to greater empathic concern and perspective taking [27]. In this case, parents whose children have been exposed to a difficult situation (such as the pandemic) present more readiness to react with empathy to their children’s cues.

The study found that some effects were gender-specific. Emotional overload in both parents increased mothers’ anxiety towards their child’s cries. This effect was not observed for fathers. Higher anxiety can lead to even greater emotional overload in mothers and may cause difficulties in everyday functioning [36]. Previous studies also indicate that anxiety in fathers does not increase overprotective behaviors toward their children [19], which may be a protective factor for the emotional overload.

Frustration towards children cries was higher in women if their partners experienced higher level of stress and lower fear of being infected. However, the frustration toward crying in men was higher if the emotional overload of their partners was higher. Women whose partners experienced higher stress can experience less support and relationship satisfaction, as reported in previous studies [37]. Furthermore, it is possible that mothers whose partners had lower fear of being infected experienced more frustration because of their partners’ disregard for the situation. This is only a tentative that should be explored in further studies.

Moreover, the low level of fear of being infected in men also raised women’s frustration because of the disregard of the pandemic situation by their partners. Men’s frustration towards their child’s crying was connected with women’s emotional overload. Their lower level of readiness to accept negative emotions in their partners, caused by a higher level of stress and an uncertain situation, could cause general frustration, including frustration directed at their child’s crying. During the COVID-19 pandemic, more than 30% of adults reported clinically meaningful symptoms of anxiety and depression [38,39], which may lead to more harsh behaviors toward one’s own children. Nevertheless, the frustration towards crying is an irritation and blaming the baby for the situation is a parent-oriented reaction. Given that it is a child-centered negative emotion, and the possibility of harsh parenting and child maltreatment [17,18,37], it is important to consider potential causes, prevent them, and facilitate appropriate vigilance towards children in support systems. However, it could also be seen that partners’ reactions to lower levels of stress provoked higher sympathy or even empathy (on the verge of statistical significance) in fathers towards their children’s crying. This underlines how important the emotions of one partner are for the functioning of the other. Such emotions can be both a risk factor for the occurrence of negative reactions towards the child, but they can be a protective factor facilitating sympathy or empathy towards a child.

Regarding the general responsiveness in women, it should be noted that that the readiness, promptness, and adequacy of their responses to child’s cues depend on the emotions of their partners. They were more responsive if their partner was less stressed by the pandemic. Although men were more responsive if their partner presented higher fear of being infected and a higher level of emotional overload. It seems that the emotional difficulties of women caused in men a greater need to care with tenderness for their child. Indeed, fathers focusing on action (which may be necessitated by pandemic and the higher level of emotional difficulties in women) leads to their being more sensitive and accepting toward their child [40].

However, only for women, there was a significant indirect effect of stress on parental responsiveness through parent-oriented frustration. As previously reported, stress is a factor that impacts effective parenting [41]. Self-oriented emotional reactions are connected with lower self-regulatory mechanisms and possible overarousal [25]. Parents who focus on their own negative emotions might not be well-fitted to the child’s situation and child’s needs [27]. Stressful situations and self-oriented responses limit empathic reactions to others. As was reported in previous studies, the more mindful parents are, the more attuned and responsive they are to their child’s needs [41].

### Limitations

This study was performed during a unique global event. Therefore, care should be used when applying the results to the general functioning of family systems. However, there is evidence that other crises had similar effects on parenting [41].

This study faces the common limitation of the sample consisting solely of volunteers, which may constitute a specific group of parents who are more interested in parenting, looking for feedback about their behaviors toward own child.

## 5. Conclusions

This study showed how the COVID-19 pandemic impacts parental reactions to their infants. It should be underlined that the experienced stress, emotional overload, or fear of being infected that are experienced in everyday life in the era of the pandemic are factors that negatively affect the condition of parents, their reactions to the child’s crying, as well as parental responsiveness. For this reason, medical, educational, care facilities, and other institutions should be more vigilant about neglect and even the maltreatment of children. The government should provide psychological support to all families to prevent the negative effects of the pandemic on entire family systems. Moreover, other studies [42] have shown that other types of crises (e.g., economic) can also impact parental emotional availability to the child. Parental responsiveness to their child’s needs will undoubtedly affect the child’s development [43,44]. Taking the above into consideration, it is crucial to provide psychological support to families.

An important step would be to expand the sample group, which would provide a broader view of the COVID-19 pandemic in Poland. The research should also be integrated with that conducted in other countries to obtain a more complete picture of the family situation. Further studies should focus also on the effects of the pandemic on child emotional and psychosocial development and also on parents’ psychological conditions.

## Figures and Tables

**Figure 1 ijerph-19-16883-f001:**
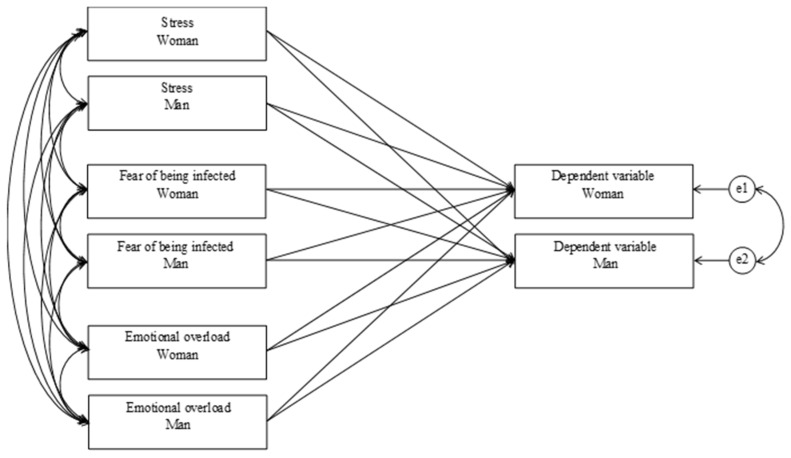
The APIMs tested for six different dependent variables: amused, anxiety, annoyance, sympathy, empathy, and parental responsiveness.

**Figure 2 ijerph-19-16883-f002:**
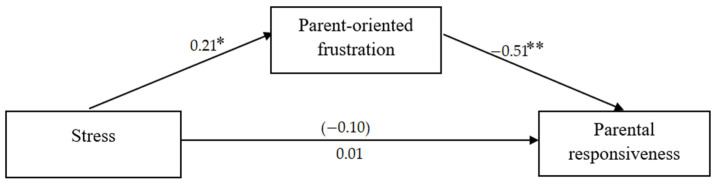
Mediation model for women (standardized coefficients are reported; total effect in parenthesis). Note. * *p* < 0.05, ** *p* < 0.01.

**Figure 3 ijerph-19-16883-f003:**
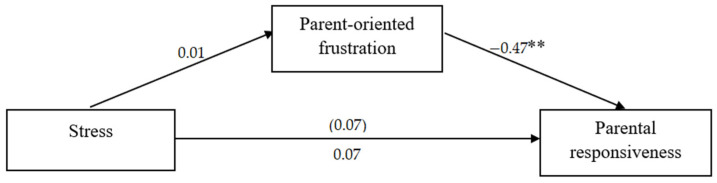
Mediation model for men (standardized coefficients are reported; total effect in parenthesis). Note. ** *p* < 0.01.

**Table 1 ijerph-19-16883-t001:** Mean scores and standard deviations (*SD*), percentages, and Pearson correlation coefficients between variables.

Variable	Mean (*SD*)	1.	2.	3.	4.	5.	6.	7.	8.	9.	10.	11.	12.	13.	14.	15.	16.	17.
Women																		
1. Stress	4.36 (2.54)	—																
2. Fear of being infected	3.75 (3.18)	0.32 **	—															
3. Emotional overload	4.68 (3.35)	0.36 **	0.63 **	—														
4. Amusement	5.03 (1.78)	−0.09	−0.08	−0.10	—													
5. Parent-oriented anxiety	9.60 (2.92)	0.05	0.11	0.25 **	−0.14	—												
6. Parent-oriented frustration	7.91 (2.63)	0.22 *	−0.06	0.15 ^†^	0.11	0.38 **	—											
7. Child-oriented sympathy	14.72 (2.79)	0.01	0.07	0.06	−0.32 **	0.21 *	−0.37 **	—										
8. Child-oriented empathy	18.67 (1.52)	0.10	0.09	0.00	−0.15 ^†^	0.11	−0.28 **	0.52 **	—									
9. Parental responsiveness	82.20 (5.56)	−0.10	−0.05	−0.12	−0.12	−0.04	−0.51 **	0.34 **	0.36 **	—								
Men																		
10. Stress	4.72 (2.83)	0.26 **	0.12	0.05	−0.05	−0.01	0.07	−0.16 ^†^	−0.05	0.14	—							
11. Fear of being infected	2.74 (2.73)	0.17 ^†^	0.55 **	0.31 **	−0.05	0.09	−0.07	0.19 *	0.13	0.06	0.29 **	—						
12. Emotional overload	4.03 (3.50)	0.19 *	0.31 **	0.31 **	0.06	0.13	0.15 ^†^	0.11	0.12	−0.01	0.41 **	0.56 **	—					
13. Amusement	5.21 (1.66)	−0.05	−0.07	−0.11	0.50 **	−0.10	0.08	−0.19 *	−0.12	−0.04	0.13	−0.03	0.09	—				
14. Parent-oriented anxiety	9.51 (3.00)	0.03	0.19 *	0.24 **	0.03	0.34 **	0.23 **	0.19 *	0.09	−0.03	0.09	0.13	0.14	0.04	—			
15. Parent-oriented frustration	9.06 (3.07)	0.01	0.01	0.12	0.10	0.23 **	0.50 **	−0.17 ^†^	0.15 ^†^	−0.35 **	0.01	−0.04	0.15 ^†^	0.19 *	0.46 **	—		
16. Child-oriented sympathy	13.09 (3.10)	−0.03	0.05	−0.03	−0.17 ^†^	0.16 ^†^	−0.07	0.42 **	0.19 *	0.22 *	0.15 ^†^	0.33 **	0.17 ^†^	−0.20 *	0.31 **	−0.14	—	
17. Child-oriented empathy	18.09 (1.87)	−0.11	−0.01	−0.02	−0.22 *	0.06	−0.19 *	0.32 **	0.18 *	0.24 **	0.20 *	0.16 ^†^	0.08	−0.29 **	−0.15 ^†^	−0.41 **	0.56 **	—
18. Parental responsiveness	77.53 (7.76)	−0.16 ^†^	−0.01	0.05	−0.08	0.05	−0.20 *	0.18 *	0.05	0.25 **	0.07	0.08	−0.09	−0.17 ^†^	−0.24 **	−0.47 **	0.31 **	0.60 **

^†^*p* < 0.10. * *p* < 0.05. ** *p* < 0.01.

**Table 2 ijerph-19-16883-t002:** Results of the Actor–Partner Interdependence Models with standardized coefficient estimates.

	Amusement		Parent-Oriented Anxiety		Parent-Oriented Frustration		Child-Oriented Sympathy		Child-Oriented Empathy		Parental Responsiveness	
Predictor	*β*	Std. Error	*β*	Std. Error	*β*	Std. Error	*β*	Std. Error	*β*	Std. Error	*β*	Std. Error
**Effect on women**												
Stress woman	−0.06	0.07	−0.04	0.11	0.20 *	0.10	0.05	0.11	0.14	0.05	−0.11	0.22
Stress man	−0.06	0.06	−0.10	0.11	−0.04	0.10	−0.06	0.13	−0.18 ^†^	0.07	−0.23 *	0.29
Fear of being infected woman	0.02	0.06	−0.10	0.11	−0.22 ^†^	0.10	−0.09	0.12	0.08	0.06	−0.01	0.20
Fear of being infected man	0.02	0.06	0.06	0.12	−0.03	0.11	−0.12	0.12	−0.12	0.06	−0.14	0.29
Emotional overload woman	−0.11	0.06	0.29 *	0.10	0.19 ^†^	0.08	0.01	0.10	−0.16	0.06	−0.08	0.18
Emotional overload man	−0.12	0.05	0.22 *	0.08	0.13	0.09	−0.05	0.11	0.07	0.05	0.23 ^†^	0.26
**Effect on men**												
Stress woman	−0.08	0.06	−0.04	0.09	−0.01	0.09	−0.28 **	0.09	−0.17 ^†^	0.05	0.18 ^†^	0.18
Stress man	0.12	0.07	0.08	0.11	−0.04	0.10	0.07	0.12	0.23 *	0.06	0.18 ^†^	0.26
Fear of being infected woman	−0.10	0.07	0.04	0.11	−0.17	0.10	0.26 *	0.10	0.09	0.06	0.11	0.23
Fear of being infected man	−0.11	0.07	0.00	0.12	−0.18 ^†^	0.12	0.41 **	0.12	0.21 *	0.06	0.23 *	0.28
Emotional overload woman	0.19 ^†^	0.05	0.07	0.09	0.22 *	0.07	0.09	0.08	0.14	0.05	−0.10	0.17
Emotional overload man	0.15	0.06	0.05	0.10	0.25 *	0.09	−0.03	0.10	−0.09	0.06	0.27 *	0.25

^†^*p* < 0.10. * *p* < 0.05. ** *p* < 0.01.

**Table 3 ijerph-19-16883-t003:** Direct, indirect, and total effects in mediation models for women and men with unstandardized coefficient estimates and 95% confidence intervals.

	Direct Effect	Indirect Effect	Total Effect
Women	0.02 [−0.31; 0.36]	−0.24 [−0.46; −0.04]	−0.22 [−0.59; 0.16]
Men	0.19 [−0.23; 0.61]	−0.01 [−0.25; 0.22]	0.19 [−0.29; 0.66]

## Data Availability

The datasets generated for this study are available on request to the corresponding author.

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
