# Peer review of "The Effects of the COVID-19 Pandemic on the Parenting of Infants: A Couples Study"

_ijerph, 2022, doi:10.3390/ijerph192416883_

Round 1

Reviewer 1 Report

Thanks for providing me the opportunity to read the article. The suggestions are:

1)      Please make sure the links and all the dois are fine. For example, the link (https://osf.io/xebhg/ - accessed April 2021) did not work on my side.

2)      The research process is not clear enough. For example, how the participants were recruited?

3)      Some figures and the numbers were too small so that I cannot see them. Please adjust the font.

Reviewer 2 Report

Thank you for inviting me to review the well-presented manuscript. This is a very interesting study; and demonstrates how the pandemic has affected all aspects of our lives. I do not have significant comments on this. Please see the comments below:    1. The questionnaire and tool section and presentation have many statistical details. A statistical review is recommended before we get more consideration. I also think the authors should consider making results more interpretative than just the statistical details.  2. It would be nice if the authors could attach or link the questionnaire they have used. The current method section does not specify if the questionnaire was in a local language or in English.  This will help the researchers in the field interpret the findings.  3. Table 1- missing heading.  4.  Discussion- The discussion can be improved by comparing and contrasting the findings from similar settings. In its current form, the discussion does not do justice to the hard work the authors have done. 5. The authors have mentioned some recommendations within the discussion of each point. However, a better presentation would be to compare and contrast and summarize the recommendation for parents and children at the end of the discussion. It is also essential to acknowledge the study's limitations as standard practice.   

Round 2

Reviewer 1 Report

I think that the authors revised and improved it. Personally, I think that the minor mistakes, such as English grammar, wording, and sentence flow could be improved. Then it is fine to publish.